# Phenological Evaluation of Minority Grape Varieties in the Wine Region of Madrid as a Strategy for Adaptation to Climate Change

Francisco Emmanuel Espinosa-Roldán [1] , Andrés García-Díaz [1], Eva Raboso [1] , Julia Crespo [1] , Félix Cabello [1], Fernando Martínez de Toda [2] and Gregorio Muñoz-Organero [1],*

1 Instituto Madrileño de Investigación y Desarrollo Rural Agrario y Alimentario (IMIDRA), Finca El Encín, 28805 Alcalá de Henares, Spain; franciscoemmanuel.espinosa@madrid.org (F.E.E.-R.); andres.garcia.diaz@madrid.org (A.G.-D.); eva.raboso@madrid.org (E.R.); julia.crespo.garcia@madrid.org (J.C.); felix.cabello@madrid.org (F.C.)

2 Instituto de Ciencias de la Vid y del Vino (ICVV), Universidad de La Rioja, CSIC, Gobierno de La Rioja, c/Madre de Dios, 51, 26006 Logroño, Spain; fernando.martinezdetoda@unirioja.es

* Correspondence: gregorio.munoz@madrid.org; Tel.: +34-91-8879483

**Abstract:** In this study, a total of 34 Spanish minority varieties were studied during four seasons from 2020 to 2023, and their behavior was characterized according to their main phenological stages (bud break, bloom, veraison, and maturity) and complete cycle. We focused on the varieties prospected in the central Spanish plateau and conserved in the "El Encín" grapevine collection, aiming to identify the potential for adaptation of these varieties and for them to be considered by winegrowers as an alternative to current climate change conditions. The growing degree days required for the expression of the phenological stage, and the duration of each stage, were compiled. Characteristics of oenological interest were also monitored, such as °Brix, pH, and titratable acidity in must at the time of harvest. This study was carried out in years with atypical snowfall and cold spells (winter 2021), as well as with heat waves (summer 2022), with average temperatures 3–5 °C higher than normal and absolute maximum temperatures over 40 °C. Both cases also exceeded records of historical series (1957–2019). Veraison has been identified as the stage most susceptible to damage from high temperatures, in addition to the maturation and duration of the complete cycle. The varieties were classified into five groups according to the duration of each phenological stage (very early, early, medium, late, and very late). Some varieties with late or very late maturation and with must characteristics of 20–23 °Brix, 3–5 g/L of titratable acidity, and pH 3.5–4.5, as well as others, retained stability in their phenological periodicity and must quality. The results suggest that special attention should be given to thoroughly evaluating these varieties, and that the strategies contemplated should be considered as a feasible cultivation alternative in viticulture to mitigate the effects of climate change.

**Keywords:** phenology; climate change; veraison; minority grapevine varieties

## 1. Introduction

Climate change projections for the near future indicate the increasing presence of extreme events, implying warmer and drier thermal conditions [1]. These changes will have a significant impact on the phenology, production, must characteristics, and oenological parameters of wine [2–4]. Temperature is the most important climatic factor for wine production [4–6], as well as the interaction with other factors such as photoperiod, water availability, quality of plant material, and viticultural techniques.

Among the main effects of climate change on vines are changes in the seasonality of the phenology, causing alterations in the thermal demand of plants and the duration of the periods of bud break, bloom, veraison, and harvest. As a result, a shortening of the bud-break-to-harvest cycle has been observed in many representative varieties in the most important wine regions of the world [7–13]. However, a few decades ago, it was considered

that increased temperature conditions would benefit the production and quality of wine in specific growing regions such as Bordeaux or Champagne [14] and California [15], where advances in phenological stages may occur due to increasingly higher thermal conditions during the winter in addition to reductions in the intensity and frequency of late frosts [2,16]. As the effects of climate change have become more frequent and extreme in recent decades, the unfavorable effect on some varieties is more evident regarding growth, physiological development, and phenological development. This does not result in any improvement in the quality of the wine produced, since high concentrations of sugar are easily reached. The alcohol content, low acidity, and alterations and instability in the aromatic compounds of the wine make it difficult to determine the optimal time of harvest [17,18].

Alterations in thermal stability, caused by climate change, aggravate the effects that are already present and give rise to sudden and irreversible changes. Due to this, plants require a higher adaptation efficiency [19,20]. In Europe, an increase has been projected in the magnitude and number of impacts due to global warming, such as alterations in the global circulation of air currents, reductions in the thickness of glacier layers, an increase in ocean temperature, and ultimately continental seasonal instability [1]. In Spain, viticulture plays an important role in the agri-food industry, with significant historical, cultural, and socio-economic value, and it is an activity that sustains the country's role as one of the leaders in the production, cultivation, and genetic diversity of the *Vitis* genus worldwide [21,22].

Numerous studies have evaluated the influence of increasing temperatures on viticulture during the 20th century [2,17], with an increase between 0.5 and 1.0 °C. An increase of more than 1.5 °C is expected by the end of the 21st century [1,23]. For example, early harvests of up to 45 days are expected in some wine regions by the year 2050 [24]. This poses a problem for the designation of regions where production practices are strictly controlled [25–27].

Over time, two potential approaches to address the impact of climate change have been identified [22]. On the one hand, there is a shift towards the development of new wine regions in areas of higher latitude and altitude in Europe, which are experiencing increasingly temperate or warmer conditions suitable for vineyards and current varieties of cultivation [28,29]. There have been adjustments to pruning techniques, crop management, and pest and disease control according to these new conditions [11,13,27,30–32]. On the other hand, the regional conservation of current vineyards has been considered, through the cultivation of adapted varieties, especially native ones that show tolerance and resistance to the new environmental conditions [9,30,32,33]. In addition, it is desirable to adopt changes in viticultural and winemaking practices to adapt to the increasingly warm and unstable climate duration [3,34,35]. The adaptation of viticultural and oenological practices to meet the challenges presented by these conditions not only represents a need but also a comprehensive opportunity for traditional wine regions. This approach ensures the complete preservation of the rich historical and cultural legacy that defines the unique identity of each growing area, winery, and wine lineage. Additionally, it ensures the protection of the rooted socioeconomic environment that has been fundamental to the development and authenticity of these regions over the years.

In this context, the use of minority and adapted varieties is a means to conserve the structure and landscape of the current wine-growing areas [36]. Through the inclusion of new varieties and the diversification of production, the varietal typicity and dominant production style in the region can be enhanced, as required by the world market: 'Tempranillo', 'Cabernet Sauvignon', 'Syrah', 'Merlot', 'Chardonnay', and 'Verdejo' [37]. These varieties continue to grow in cultivation areas [38], relegating to small areas those also widely cultivated in the 20th century: 'Airén', 'Garnacha', 'Monastrell', 'Bobal', 'Cayetana Blanca', 'Palomino', and 'Pedro Ximénez'. These 13 varieties (1% of the total diversity of *Vitis vinifera* L.) represent between 70% and 90% of the vineyards in wine-growing countries. The conservation of grapevine varietal diversity in ampelographic collections facilitates a simultaneous evaluation of the adaptation of varieties to climate change in the same region

and under the same environmental conditions. This can help winegrowers in adapting current production regions to climate change using their traditional varieties [13,29,30].

In 'El Encín' (Alcalá de Henares, Madrid), the largest varietal collection in Spain is preserved, constantly referenced, and updated using morphological and molecular characterization techniques [13,21,39]. Several authors have focused on the characterization of varieties conserved within collections. Their behavior as a group has been addressed concerning the change in temperatures in historical series, and estimates have been offered of the quality of the must obtained as well as the appearance and duration of the phenological stages in collections or cultivation regions of interest. It is essential to know how temperature influences both their reproductive cycle and vegetative development, and to observe differences between varieties [4,40].

This paper studies the historical thermal characterization of 'El Encín' from 1957 to 2019 and compares the information with the period of this study (2020 to 2023). Special attention is paid to the 2021 and 2022 seasons. The first represents a year with an exceptionally atypical winter caused by the storm 'Filomena' [41,42], followed by a 'cold spell' for 11 consecutive days. The second is a year with very warm annual records: a particularly long, hot, dry summer, with 'heat waves' that broke historical records in quantity, duration, and intensity [41]. Both years are exceptionally atypical concerning the history recorded in the collection's weather station. As a result, alterations were observed in the duration of the cycles, the phenological stages, and the characteristics of the grape established for the determination of the harvest date of each variety.

Therefore, the aim of this work is to analyze the effect of temperature on the phenology, seasonality, and ripening parameters of minority grapevine varieties in the region of Madrid, as well as to identify the varieties that can be considered as crop alternatives under the current conditions of climate change. To this end, the phenological behavior of 34 minor varieties from the 'El Encín' ampelographic collection was evaluated. This provided information that allowed us to identify those of greatest interest and cultivation potential, taking into account the conditions of climate change and increased temperatures.

## 2. Materials and Methods

The present study was conducted in the center of the Iberian Peninsula, in the vine varieties collection of 'El Encín' located in Alcalá de Henares, Spain (40°310′ N, 3°170′ W, 610 m.a.s.l) in a flat river terrace of the Henares River. The climate is semi-arid Mediterranean with a xeric soil moisture regime, less than 400 mm of rainfall per year, and an average temperature of around 14° C (1957–2019). A total of 34 minority vine varieties, 16 white and 18 red, were studied, all of which are located and conserved in El Encín (Table 1). 'Tempranillo' (red), 'Garnacha tinta' (red), and 'Airén' (white) were used as reference varieties due to their wide cultivation area at regional and national levels, as well as 'Malvar' (white) because of its regional importance. The origin of the plant material is the result of joint surveys between farmers and research groups in enology and viticulture on different wine-growing regions throughout Spain during the last two decades.

A study of the phenology of the 34 varieties was carried out using the BBCH scale [43] in the following stages: bud break (BBCH 03), flowering 50% (BBCH 65), veraison (BBCH 83), and maturity (BBCH 89). These four stages were selected because of their agronomical relevance and the results of previous studies [13]. The duration of each phenological stage was calculated as the period from the onset of the stage to the onset of the following stage, taking into account all stages in the BBCH scale. For each variety, the duration of the phenological stages in Julian days (JDs) was estimated, as well as the growing degrees days (GDDs) required for the onset of the phenophase [44]. In addition, the total duration of the vegetative–reproductive (CC) cycle (from bud break to maturation) and the total GDDs between these stages were calculated (Supplementary Table S2).

**Table 1.** Minority varieties studied and region where the original plant material was prospected.

| Variety | Skin Color | Prospected Region |
|---|---|---|
| Airén | white | Castilla-La Mancha |
| Albillo del Pozo | white | Castilla-La Mancha |
| Aurea | white | Castilla y León |
| Azargón | red | Castilla-La Mancha |
| Benedicto | red | Galicia |
| Cadrete | red | Aragón |
| Castellana Blanca | white | Castilla y León |
| Crepa | red | Castilla-La Mancha |
| Folgasao | white | Galicia |
| Garnacha Tinta | red | Andalucía |
| Granadera | red | Castilla-La Mancha |
| Hebén | white | Castilla y León |
| Jarrosuelto | white | Castilla-La Mancha |
| Listan Prieto | red | Canarias |
| Lucomol | white | Castilla-La Mancha |
| Malvar | white | Castilla-La Mancha |
| Marfileña | white | Castilla-La Mancha |
| Montonera | white | Castilla-La Mancha |
| Morate | red | Navarra |
| Pintada | white | Castilla-La Mancha |
| Rayada Melonera | red | Comunidad de Madrid |
| Rubeliza | red | Castilla-La Mancha |
| Salvador | white | Castilla y León |
| Sanguina | red | Cataluña |
| Tazazonal | red | Navarra |
| Tempranillo | red | Navarra |
| Terriza | red | Castilla-La Mancha, Madrid, Navarra |
| Tinto Bastardo | red | Castilla-La Mancha |
| Tinto de Navalcarnero | red | Castilla y León |
| Tinto Fragoso | red | Castilla-La Mancha |
| Tortozón | white | Extremadura |
| Tortozona Tinta | red | Castilla-La Mancha |
| Verdejo Serrano | white | Castilla y León |
| Zurieles | white | Castilla-La Mancha |

The vines were grafted on Richter 110 in 2002 and trained in a unilateral cordon system with eight to ten buds per vine. They were established in a conventional planting system designed with 0.90 m between plants and 2.50 m between rows, with a planting density of 4444 vines/ha. Phenological monitoring was performed on 10 plants per variety, and the same cultural management practices were applied to the collection: winter pruning, green pruning, soil tillage management, and minimal application of phytosanitary products. No trimming of shoots or bunches was performed.

The climate data were recorded at the weather station located in El Encín, which is certified by the Spanish Meteorological Agency. Daily records of mean, minimum, and maximum temperature (°C) were collected in 10-min series during the four years of the study. Average daily temperatures were calculated using 142 daily air temperature (°C) records. The GDDs were calculated by subtracting the quotient obtained by dividing by two the sum of the maximum daily temperature (Tmax) and the minimum daily temperature (Tmin) [5,45,46] from the base temperature or threshold temperature of the plant (10 °C) [44,45,47]. Finally, each year of the study was characterized. The average, maximum, and minimum monthly temperatures of the historical series from 1957 to 2019 recorded at the 'El Encín' meteorological station were compared with the values from the years in which the study was conducted (2020 to 2023). Special attention was paid to the 2021 season, characterized by the presence of snowfall followed by an atypical cold spell, and to the 2022 season, which recorded one of the warmest and driest summers in Spanish history,

in addition to the presence of longer-lasting and more intense heat waves in recent years. This information may help to explain the phenological behavior of the varieties, mainly during bud break and maturation, as well as their response to increasingly frequent extreme and atypical meteorological events such as those that occurred during the study period.

Harvest dates were determined based on total soluble solid accumulation (°Brix) in berries: 20–21 °Brix for white varieties and 21–22 °Brix for red varieties. Indicators of oenological quality were obtained at harvest time: °Brix, pH, and titratable acidity. The °Brix to track ripening and harvest time was calculated with a digital refractometer (PR-101 Palette, ATAGO®, Tokyo, Japan). The pH was obtained with a laboratory LPG meter with temperature and pH electrodes (Sension+ PH31, HACH®, Loveland, CO, USA), and the titratable acidity value (g/L of tartaric acid) was determined through titration with 0.1 N sodium hydroxide, taking the sample to pH 7 with bromothymol blue as an indicator for red varieties and phenolphthalein for white varieties [48].

To contrast the hypotheses, the following statistical analyses were carried out: (1) the correlation between the study of phenological variables and the GDDs was estimated by calculating the non-parametric Kendall tau-b statistic (appropriate for small samples with numerous values in the same range and non-normal distributions); (2) the K-means procedure was used to sort the different grape varieties studied into groups that were as homogeneous as possible based on the duration of the different phenological states (bud break, bloom, veraison, ripening, and complete cycle) and thus classify them into very early (VE), early (E), medium (M), late (L), and very late (VL) varieties. SPSS statistical software version 23 [49] was used for the analyses.

## 3. Results

### 3.1. Temperature Assessment

The average monthly temperatures were higher during the study period compared to the average monthly temperatures of the historical series (1957–2019), which highlights trends of thermal increase and extreme seasonal variations caused by climate change. The average temperature during the entire period of reproductive vegetative growth was higher in all the years of study (2020–2023) compared to those recorded historically. The monthly absolute minimum and maximum temperatures became more extreme and tended to be higher (Figure 1a). Winters became less cold and shorter with higher temperatures than normal at the end of the season. On the other hand, summer tended to be prolonged, starting earlier and even ending later than usual. From June to August 2022, the average temperatures recorded were 3–5 °C higher than those in historical series. On the other hand, winters tended to be slightly warmer, with periods between 2 and 3 °C higher than the historical average but with an increasingly frequent presence of unusual events such as storms, snowfall, and cold spells, such as those that occurred in January 2021 (Figure 1b).

Exceptional weather conditions were experienced during January 2021, marking a milestone in the historical temperature records. The combination of the 'Filomena' snowfall and the subsequent cold wave produced a singular phenomenon, with an absolute minimum temperature of −13.7 °C that contrasted significantly with the historical average of −5.5 °C for this time of year. The absolute minimum of 0.6 °C recorded in February of the same year was also higher compared to previous years, which usually recorded figures below −1.0 °C. Throughout the summer of 2021, average temperatures remained slightly below usual standards between June and September, with June to August being the warmest months, with records of 24.8 °C and 25.6 °C, respectively.

The year 2022 was the warmest of the periods studied. At the end of winter (March), higher absolute minimum temperatures than usual were observed; a minimum temperature of −1.0 °C was recorded, while in other years temperatures fell below −2.0 °C. The absolute maximum temperatures of January and February remained constant around 25.0 °C during the four years, with a slight decrease in March (18.3 °C). From May, a notable increase in temperatures began, extending until October, with the absolute maximum and average monthly maximum temperature up to 3.0 °C higher than in previous years. Persistent heat

waves during the summer led to exceptional absolute maximum temperatures, reaching 40.0 °C in June and July and surpassing 39.1 °C in August. The annual average temperature for this period was 15.9 °C, representing an increase of 2.0 °C compared to the historical average recorded between 1957 and 2019 (13.9 °C).

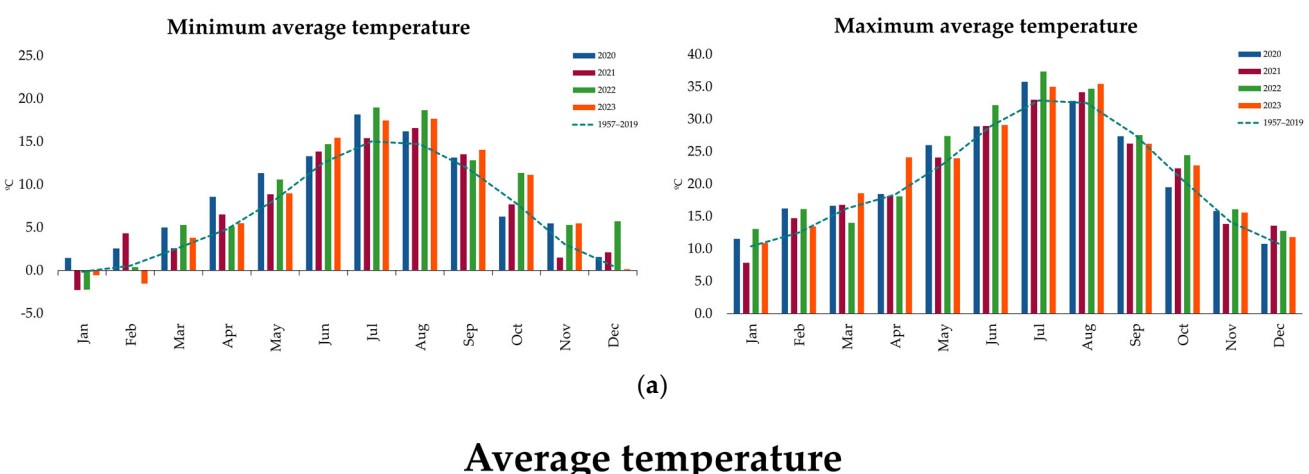

(**a**)

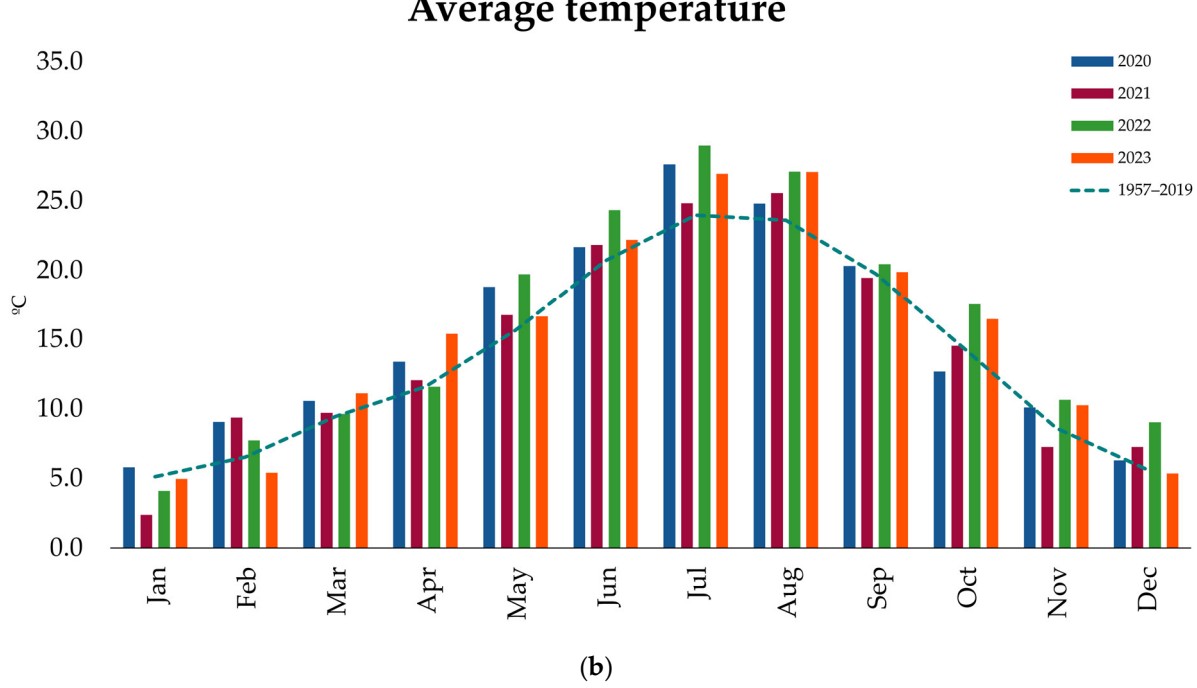

(**b**)

**Figure 1.** (**a**) Monthly average minimum temperature (left) and monthly average maximum temperature (right) during the study (2020–2023). (**b**) Monthly mean temperature during each month of the study and historical monthly mean temperature (1957–2019).

From March to December 2022, monthly average temperatures exceeded both those recorded in previous years and the historical average. May and October stood out with records up to 3.0 °C higher than other years. The months of June, July, August, and September maintained monthly average temperatures above 20 °C, with absolute maximums exceeding 40.0 °C. July and August were consolidated as the hottest period of the year, with average temperatures above 27 °C. In July and August 2023, the weather conditions were similar to those of the previous year, with absolute maximums again exceeding 40.0 °C and average temperatures above 27 °C. There was an upward trend in absolute maximum temperatures during spring and autumn, as well as in monthly average temperatures from March to November, with increases of up to 1.0 °C compared to previous years and even exceeding the historical record of 'El Encín' by 2.0 °C.

*3.2. Phenology*

Differences were observed in the controls and minority varieties in the duration of the phenological stages (bud break, flowering, veraison, and maturation) and complete cycle (Table 2) and in the length of the full cycle (BBCH 83, Figure 2). In 2021, the veraison period (BBCH 83) lasted approximately 12.06 days, which was 4 to 5 days longer than in the other years of the study. During 2022 and 2023, the durations were 7.41 and 7.38 days, respectively (Supplementary Table S2). The veraison stage of the white control varieties 'Airén' and 'Malvar' was rated as very late and medium, respectively. On the other hand, the red control varieties 'Garnacha tinta' and 'Tempranillo' were rated with an early and medium veraison period, respectively (Supplementary Tables S2 and S3).

In addition to the duration of the final phenological stages, the length of the vegetative–reproductive cycle from bud break (BBCH 03) to maturity (BBCH 89) is determined by the earliness of onset and duration of the previous phenological stages (Supplementary Tables S2 and S3). In 2022, the length of the complete cycle was the shortest of the four study years, with 150.12 days; meanwhile, 2020 and 2021 were years in which the cycle took 172.73 and 163.32 days to complete, respectively; these were the two warmest years during the study period.

Statistical analysis provided an estimate of duration (JDs) and degrees of days required (GDDs) for each phenological stage during the study period. During 2022, the duration of the bud break period (BBCH 03) and the accumulation of GDDs were the highest of the four years; the typical spring-like temperatures recorded in the last weeks of winter led to very early bud break in varieties that usually budded early ('Pintada', 'Rubeliza', 'Jarrosuelto', and 'Sanguina') or medium ('Listán Prieto') and even late ('Airén', 'Cadrete', 'Morate', and 'Tinto de Navalcarnero') seasonality. Another group of varieties sprouted early, usually with late ('Benedicto', 'Malvar', and 'Rayada Melonera') or very late ('Castellana Blanca') seasonality. The flowering period (BBCH 65) throughout the study was the most stable and suffered the fewest alterations due to the thermal variations present during 2021 and 2022, with an approximate duration of between 4 and 8 days depending on each variety and considering the flowering period of the control varieties. This is a period that can be considered normal (Supplementary Tables S2 and S3).

Veraison was the phenological stage in which the effect of thermal variations was more critical in two of the four years of the study (Figure 2). In the 2021 season, the veraison period was extended by 12.06 days and it required 215.16 GDDs to initiate the color change that gave rise to the beginning of the ripening process. The reasons for these high values compared to other seasons of the study could be (1) the loss of productive buds due to the low temperatures caused by the 'Filomena' snowfall in January 2021; (2) cooler spring temperatures than usual; and (3) the longer time until the bud stage, resulting in a higher GDD requirement to sprout. A high percentage of the buds that sprouted only developed vegetation and presented a notable reduction in fertility.

The varieties were classified into five clusters at the veraison stage, highlighting the reference variety 'Airén', with stability over the years in the group of varieties in which the veraison stage occurred very late. 'Terriza', 'Folgasao', and 'Tinto de Navalcarnero' are usually grouped as late (L) or very late (VL) varieties. The group of varieties with a medium veraison period (M) varies over the years of study; 2021 was the season with the fewest number of varieties with a medium veraison period, but there were also few varieties in the other years ('Tempranillo' and 'Tinto Fragoso'). The varieties that showed an early veraison (E) period and were stable over the years were 'Garnacha tinta' and 'Montonera'. Other varieties that have reached a certain stability and fluctuate between an early (E) or very early (VE) veraison period are 'Listán' Prieto', 'Rayada Melonera', and 'Tazazonal'.

**Table 2.** Duration of the main phenological stages (JDs), standard deviation (SD), and GDDs in each year of study.

| Phenology Stage | Year | JD | SD | GDD | SD | Cases |
|---|---|---|---|---|---|---|
| Bud break | 2020 | 6.36 | 3.16 | 5.65 | 7.47 | 23 |
|  | 2021 | 6.24 | 4.13 | 15.46 | 8.78 | 34 |
|  | 2022 | 8.62 | 5.53 | 20.14 | 7.58 | 34 |
|  | 2023 | 4.41 | 1.92 | 14.37 | 10.83 | 34 |
| Bloom | 2020 | 5.57 | 1.16 | 61.08 | 16.39 | 23 |
|  | 2021 | 4.21 | 1.65 | 52.89 | 20.58 | 34 |
|  | 2022 | 6.94 | 1.94 | 75.92 | 22.32 | 34 |
|  | 2023 | 8.55 | 3.29 | 61.09 | 22.38 | 34 |
| Veraison | 2020 | 8.50 | 7.82 | 135.79 | 128.02 | 26 |
|  | 2021 | 11.62 | 6.39 | 178.29 | 105.68 | 34 |
|  | 2022 | 7.41 | 4.06 | 129.56 | 66.68 | 34 |
|  | 2023 | 7.38 | 3.28 | 126.66 | 56.61 | 34 |
| Harvest | 2020 | 28.42 | 13.23 | 370.44 | 164.76 | 26 |
|  | 2021 | 22.41 | 9.75 | 303.30 | 140.74 | 34 |
|  | 2022 | 25.68 | 9.51 | 385.08 | 130.08 | 34 |
|  | 2023 | 30.68 | 13.97 | 444.39 | 166.72 | 34 |
| Complete cycle | 2020 | 172.73 | 11.64 | 1800.47 | 113.83 | 26 |
|  | 2021 | 163.32 | 12.08 | 1641.72 | 126.01 | 34 |
|  | 2022 | 150.12 | 12.97 | 1846.05 | 177.37 | 34 |
|  | 2023 | 160.21 | 16.72 | 1798.96 | 197.75 | 34 |

JD: duration of the phenological stages in Julian days; GDD: degrees day of growth: SD: standard deviation.

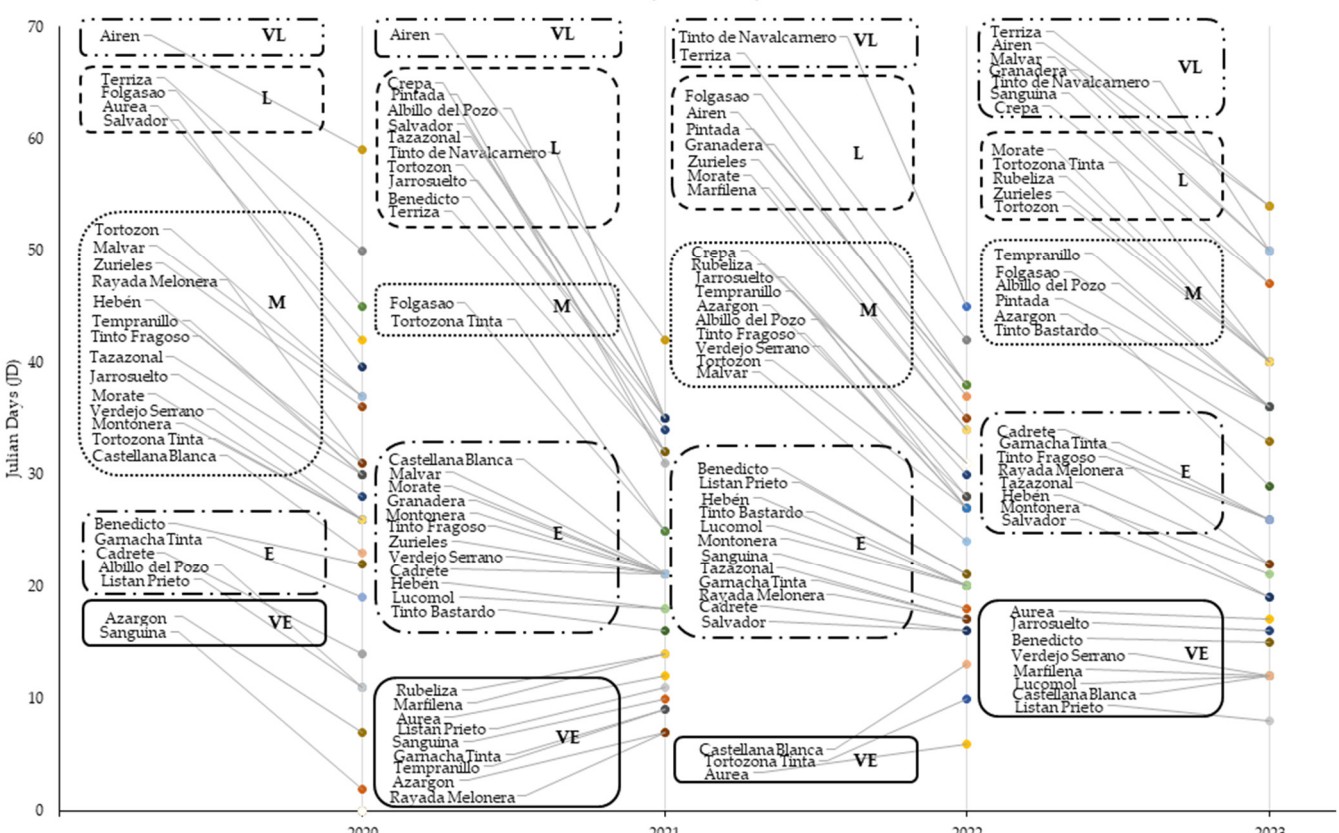

**Figure 2.** Groups of varieties according to veraison (BBCH 83) harvest period in each year of study (2020 to 2023) in "El Encín" (Madrid): VE: very early; E: early; M: medium; L: late; VL: very late. There are missing data for some varieties in 2020 (JDs = 0).

### 3.3. Correlation of GDDs and JD of Phenology Stages

A highly significant relationship has been identified between climatic conditions and their effects on the phenology of the minority varieties evaluated. Kendall's tau-b correlation test yielded highly significant correlations between the degree days of growth (GDDs) and duration (JDs) of the four phenological stages (bud break, bloom, veraison, and maturity) and the complete cycle of the varieties. The relationship is highly significant for the final phenological stages of the cycle (veraison and maturation) and the complete cycle (Table 3). The veraison is the phenological stage that showed a high correlation between GDDs and JDs during the 4 years of this study. In the case of ripening, the correlation was highly significant in the 2020, 2022, and 2023 seasons.

**Table 3.** Correlation coefficients between the duration of the phenological stages in Julian days (JDs) and the growing degree days (GDDs) in the same stage for 34 varieties sampled during 2020, 2021, 2022, and 2023.

| | **Kendall Tau-b Correlation** | **2020** | **2021** | **2022** | **2023** |
|---|---|---|---|---|---|
| Bud break | Coefficient | −0.111 | **0.603 \*\*** | **0.434 \*\*** | **0.362 \*** |
| | Bilateral significance | 0.525 | 0.000 | 0.001 | 0.011 |
| | N | 22 | 34 | 34 | 34 |
| Bloom | Coefficient | 0.006 | **0.917 \*\*** | **0.788 \*\*** | **0.788 \*\*** |
| | Bilateral significance | 0.973 | 0.000 | 0.000 | 0.000 |
| | N | 23 | 34 | 34 | 33 |
| Veraison | Coefficient | **0.894 \*\*** | **0.854 \*\*** | **0.824 \*\*** | **0.909 \*\*** |
| | Bilateral significance | <0.001 | 0.000 | 0.000 | 0.000 |
| | N | 26 | 34 | 34 | 34 |
| Harvest | Coefficient | **0.899 \*\*** | **0.791 \*\*** | **0.827 \*\*** | **0.864 \*\*** |
| | Bilateral significance | <0.001 | 0.000 | 0.000 | 0.000 |
| | N | 26 | 34 | 34 | 34 |
| Complete cycle | Coefficient | **0.838 \*\*** | **0.728 \*\*** | **0.685 \*\*** | **0.922 \*\*** |
| | Bilateral significance | <0.001 | 0.000 | 0.000 | 0.000 |
| | N | 26 | 34 | 34 | 34 |

Highly significant correlations at the 0.01 level (two-sided) are marked with \*\*, and highly significant correlations at the 0.05 level (two-sided) are marked with \* and highlighted in bold. $p < 0.01$ was defined for Kendall's Tau-b probabilities. JD: duration of the phenological stage, in days; GDD: accumulated degree days for each phenological stage; N: number of cases analyzed (in 2020 the value of N < 34 because phenological values of some varieties were not collected).

The complete cycle turns out to have a highly significant correlation during the first and last years of this study (2020 and 2023); meanwhile, during 2021 and 2022 it registers a medium–high correlation at 0.01. In 2021, the intensity of the relationship between the GDDs and the JDs of flowering and veraison is high, with a significance level of 0.01. In 2022, the correlation is highly significant at 0.01 for the veraison and ripening periods. The correlations are highly significant for veraison, ripening, and full cycle at 0.01 in 2023, and are similar to those observed in 2022.

### 3.4. Composition of the Must at the Time of Harvest

After monitoring the ripening process and determining the harvest time, it was possible to classify all the varieties within five clusters (Table 4 and Supplementary Table S3) according to the degree of earliness and determine the average characteristics of the must obtained in each year (Table 4). During the four years of this study, the very early (VE), early (E), and medium (M) varieties presented must quality values within those established in the methodology: °Brix at the time of harvest between 20 and 21 °Brix, pH values around 3.5, and average titratable acidity between 4.5 and 5.0 g/L.

**Table 4.** Average must parameters in minority varieties at the time of harvest during the study period (2020–2023).

| Maturation Cluster (BBCH 89) | Year | °Brix Means | | pH Means | | Titratable Acidity Means | |
|---|---|---|---|---|---|---|---|
| | | Average | SD [1] | Average | SD [1] | Average | SD [1] |
| Very early | 2020 | 21.00 | . | 3.02 | . | 6.06 | . |
| | 2021 | 22.04 | 1.09 | 3.66 | 0.28 | 4.86 | 2.22 |
| | 2022 | 22.03 | 1.40 | 3.35 | 0.20 | 6.35 | 0.79 |
| | 2023 | 22.89 | 1.37 | 3.49 | 0.26 | 5.52 | 1.30 |
| Early | 2020 | 22.60 | 0.32 | 3.41 | 0.16 | 4.31 | 1.05 |
| | 2021 | 22.37 | 1.62 | 3.55 | 0.18 | 4.84 | 1.12 |
| | 2022 | 22.46 | 1.99 | 3.51 | 0.24 | 5.24 | 1.48 |
| | 2023 | 22.54 | 1.25 | 3.56 | 0.20 | 4.88 | 1.39 |
| Medium | 2020 | 22.44 | 1.24 | 3.59 | 0.28 | 4.55 | 1.01 |
| | 2021 | 21.80 | 2.40 | 3.33 | 0.31 | 6.95 | 1.48 |
| | 2022 | 22.09 | 2.65 | 3.58 | 0.17 | 4.35 | 1.07 |
| | 2023 | 21.73 | 2.06 | 3.68 | 0.10 | 4.05 | 0.49 |
| Late | 2020 | 21.40 | 2.80 | 3.60 | 0.09 | 4.25 | 0.87 |
| | 2021 | 21.91 | 2.96 | 3.61 | 0.18 | 4.43 | 0.51 |
| | 2022 | 21.20 | 1.27 | 3.61 | 0.28 | 4.24 | 1.37 |
| | 2023 | 22.36 | 1.12 | 3.53 | 0.16 | 3.95 | 0.64 |
| Very late | 2020 | 20.30 | . | 3.79 | . | 4.50 | . |
| | 2021 | 21.50 | . | 3.80 | . | 3.50 | . |
| | 2022 | 18.48 | 2.02 | 3.67 | 0.18 | 4.05 | 0.07 |
| | 2023 | 20.06 | 2.12 | 3.42 | 0.25 | 4.97 | 1.93 |

[1]: Standard deviation (SD); (.): absence of data for one or two varieties during the study year.

Among the control varieties, the phenology and ripening of 'Garnacha tinta' remained stable during this study. This variety ripened early (E). The effects of the damage caused by the cold of the winter of 2021 slightly altered the ripening of the variety, which became very early (VE) this year. On the other hand, 'Tempranillo' was part of the cluster of medium-ripening (M) varieties throughout the study period, except in 2021 when it was part of the very-early-ripening (VE) varieties due to how early flowering occurred in the few bunches that the variety produced. On the other hand, 'Airén' was the most stable of the four control varieties in each year of the study. It was part of the very-late-ripening (VL) varieties cluster, except in 2022, when its ripening was accelerated and it became part of the late varieties (L) cluster.

Varieties such as 'Terriza' and 'Tinto de Navalcarnero' stand out, with late ripening (L) in 2020 and 2021 and very late ripening (VL) in 2022 and 2023; these were the last two years of the study and had higher summer temperatures (Supplementary Tables S2 and S3). Despite ripening slower, they managed to accumulate the levels of total acidity, pH, and °Brix necessary for harvesting under good winemaking conditions. In addition, varieties such as 'Tortozon' and 'Morate' achieved the same beneficial results in their cycle, showing medium ripening (M) and late ripening (L) under the same conditions.

Other varieties reduced their ripening period under the influence of high temperatures, such as 'Áurea', 'Folgasao', and 'Tazazonal'. Some also tended to ripen for longer, sometimes with a sudden stop in the accumulation of °Brix, for up to 3 weeks in the presence of temperatures above 38 °C that occurred during the heat waves of the summer of 2022 and 2023. Varieties such as 'Azargón, 'Cadrete', and 'Listán Prieto' stand out for their low titratable acidity values at the time of harvest in 2022 and 2023, which were the warmest and driest of the study period (Supplementary Tables S1 and S3).

## 4. Discussion

By considering the historical temperature series and the variability in thermal conditions during the study period (2020–2023), it was possible to determine the phenological

behavior of the varieties even under extreme conditions, including cold stress during winter (2021) and high temperatures and repeated heat waves during summer (2022). The correlations between the values of degree days (GDDs) and the duration of phenological stages (JDs) throughout the four years of this study are mainly significant in the late stages of the cycle, particularly during the summer season when veraison and maturation processes mainly take place [13]. The bud break and flowering stages occur in very limited and narrow periods in each variety and are triggered by the onset of spring temperatures and the metabolic awakening of the plant. The effects that higher temperatures have had on the earlier bud break are mainly related to the susceptibility of buds with green shoots to late frosts. These frosts are becoming less frequent in the central plateau of the Iberian Peninsula, but they can still cause problems in terms of crop loss and annual production stability. The same is seen with flowering given that the varieties have a significantly reduced period of time in which pollination can occur or present an imbalance between flowering and fruit setting in the same bunch, as was the case for 'Malvar' and 'Benedicto' in the warmest years (2022 and 2023).

Thermal variability in the early stages of growth has an influence on the phenological behavior of plants, mainly when the base temperature (10 °C) is reached for the start of physiological activity [50]. An alteration in the initial phenological stages was observed in 2021 due to the cold conditions present during January of that year. The buds defined for production were mostly affected by low temperatures, so bud break was observed in the basal buds of the spurs or even in the dormant buds. The varieties least affected by cold damage in the winter of 2021 were 'Benedicto', 'Castellana blanca', 'Morate', 'Salvador', 'Tempranillo', and 'Tinto de Navalcarnero'. All of them sprouted and generated fertile buds (although with a certain decrease in productivity) at the end of the cycle. The effect of low temperatures on the buds of spurs was observed in the rest of the varieties with different degrees of intensity affecting sprouting and, consequently, productivity. The most affected varieties were 'Jarrosuelto' and 'Sanguina', which completely lost their productive buds that year. Other varieties that suffered a drastic decrease in shoots from productive buds were 'Àurea', 'Crepa', 'Granadera', 'Terriza', 'Rayada melonera', 'Marfileña', 'Montonera', 'Verdejo Serrano', and 'Zurieles'.

Some studies show an increase in temperatures during the period of growth and ripening of the vine, between 1 °C and 2.2 °C in northern Spain [51,52], up to 2.3 °C in the Veneto region in Italy (1964–2009) [53], and between 3 °C and 4 °C in Madrid, Spain, from 1957 to 2021 [13]. As can be observed, the absolute maximum temperatures easily reached 40.0 °C in 2022 and in 2023, and they exceed that level more frequently than usual; this is a temperature that had not occurred in previous years at this frequency. However, although these were extraordinary events, they are now considered summer temperatures that are becoming increasingly common. It is worrying that these extreme temperatures are accentuated by the presence of increasingly longer summers and more extreme droughts in rural regions in recent years, coupled with increasingly scarce and erratically seasonal rainfall [20]. The implications of these extreme thermal conditions of drought during the summer, with very high absolute maximum temperatures, in addition to the fact that the average temperatures in summer during recent years have increased by up to 3.0 °C compared to the historical series (1957–2019), will evidently have implications for the seasonality of the phenology of widely cultivated varieties and other minorities. In some cases, sprouting will come earlier, putting the plant's buds at risk of experiencing late frosts (which are increasingly less frequent). The success of flowering and fruit setting is also conditioned as a result of the reduction in the duration of these stages and the fact that they occur under abrasive temperatures. Therefore, this will cause alterations in the productivity and quality of the bunch during veraison, in addition to the reduction in the periods of these stages, the risk of burns, and a loss of productive buds. Ripening and the complete cycle (CC) will be affected by the ripening process being stopped or by burns on the clusters, as well as changes in the seasonality of the varieties' harvest [36,54], which arouses interest year by year because harvest comes earlier in the most important wine-growing regions

in the country. The duration of the complete cycle from bud break to maturation has also been severely affected.

The reduction is alarming because earlier harvests have been recorded in varieties whose maturation is medium or late and with early or medium bud break. In 2022, the duration of the complete cycle, from bud break to maturation (BBCH 03 to BBCH 89), was reduced by 22 and 13 days, respectively, compared to 2021 and 2020 [55,56]. This reduction is extremely high, as it has occurred in consecutive years and is increasingly advancing, generating harvests with oenological imbalances due to the accumulation of sugars and thus a probable high alcohol content, unstable yields, high pH, and low acidity. This confirms the predictions and experiences recorded by other authors [11,13,29,52] and the current records in other denominations of origin, where the harvest dates have started 15 and up to 21 days in advance of the usual dates in recent years.

The effect that high temperatures have on the phenology and quality of the must of varieties [4,13,33,56] such as 'Airén' is evident; these varieties usually have very late (VL) periods of veraison and ripening with very good values of °Brix, acidity, and pH. However, in hot years such as 2022, the quality of the must tends to be compromised because the ripening period is extended and the risk that the accumulation of these parameters and other aromatic compounds (not considered in this work) is stopped increases.

Very late varieties showed greater variability in must quality parameters due to climate instability during the 4 years of this study. This effect was higher in 2021 due to the low production in each variety resulting from cold damage at the original fruiting points determined by annual pruning. This also occurred in 2022 because of high temperatures affecting the seasonality of the ripeness of most of the varieties, according to the oenological parameter determined to establish the time of harvest (°Brix accumulation). High temperatures, drought, and constant heat waves during the summer of 2022 accelerated the ripening process of many varieties. 'Malvar', one of the control varieties, presented a very irregular phenology throughout the study, with problems in the duration of the phenological stages, in the setting of bunches, in the flowering dates, and finally in ripening. In 2020 and 2022, it was part of the cluster of medium-ripening varieties (M), with the latter year being noticeably warmer and drier than the first year of the study. During 2021, as with many varieties affected by the very low temperatures, ripening occurred slightly earlier than expected this year and it was part of the early-maturing varieties (VE). Finally, in 2023, it was part of the cluster of very-late-ripening varieties (VL). However, it is a variety that is known to be highly sensitive to the climatic conditions present in the region, and the variability in its ripening season has already been debated. Historically, it is known as a late-ripening variety [57], although contemporary studies classify it as an early-ripening variety [58,59]. Therefore, the ripening trend of this variety was the most typical during the first years of this study; however, it did have a tendency to lengthen its ripening process, according to the results obtained.

The cold snap lasting 11 days after the snowfall froze the buds, and a high percentage of them did not sprout during the spring.

Medium-ripening (M) varieties with good parameters of acidity, pH, and °Brix at the time of harvest, such as 'Rayada Melonera', 'Jarrosuelto', 'Tortozona Tinta', and 'Morate', as well as 'Airén', can be considered profitable options, given that they achieve good standards in terms of must quality and were varieties with good yields in this period (values not included in this work). Other varieties that can be considered viable given their characteristics and performance under stressful conditions of heat (2022) and late maturation (Ta) are 'Terriza' and 'Sanguina'. Together with the previous varieties, these may present greater chances of success when grown in increasingly extreme and changing climatic conditions such as those observed in recent years.

## 5. Conclusions

Climate change is a phenomenon that increasingly affects the environment of vines worldwide, and its effects are increasingly evident with the occurrence of extreme events

such as more intense and frequent heat waves and cold spells, and a loss of the seasonality of these events. High temperatures during the vine cultivation cycle affect its phenology, mainly the veraison stage, with notable consequences for the final stages, such as problems in fruit setting, a shortening of the ripening period, and in some cases the complete termination of the ripening process due to high temperatures. These extreme conditions also have an effect on the quality of the must of traditionally cultivated varieties, in addition to shortening the cultivation cycle and requiring modifications to cultivation tasks during the winter and harvest dates and potentially modifications to the winemaking process according to the maturation conditions. It was possible to group all the varieties in each of the years into five groups according to the time of occurrence of each of the phenological stages studied (bud break, bloom, veraison, maturity, and complete cycle): very early (VE), early (E), medium (M), late (L), and very late (VL). The varieties that were most affected by heat waves are 'Tempranillo', 'Malvar', 'Listán Prieto', 'Azargón', and 'Cadrete'. However, other varieties have been found that exhibit resistance patterns to these conditions with good oenological values at the time of harvest, such as 'Airén', 'Tinto de Navalcarnero', 'Tortozona Tinta', 'Tortozón', and 'Morate', in addition to varieties such as 'Garnacha Tinta' that maintain acceptable productive conditions under the observed changes according to established winemaking standards. The following varieties can be considered those with possibilities for good performance under the current climatic conditions, depending on their response in terms of phenology and production quality: 'Airén', 'Morate', 'Rayada Melonera', 'Tortozona Tinta', and 'Tinto de Navalcarnero'.

**Supplementary Materials:** The following supporting information can be downloaded at https://www.mdpi.com/article/10.3390/horticulturae10040353/s1: Table S1: Composition of the must in minority varieties and controls at the time of harvest in the four study seasons (2020 to 2023); Table S2: Phenology of the varieties (onset, end, and duration) of each phenological state, including the complete cycle of all the varieties studied from 2020 to 2023. Table S3: Classification of varieties into homogeneous groups using the K-means procedure, according to their level of precocity in each phenological stage for each year of study (2020 to 2023).

**Author Contributions:** Conceptualization, G.M.-O., F.M.d.T., F.C. and F.E.E.-R.; formal analysis, F.E.E.-R., F.M.d.T., F.C. and G.M.-O.; investigation, F.E.E.-R.; resources, F.E.E.-R., J.C., A.G.-D. and G.M.-O.; data curation, F.E.E.-R., E.R. and J.C.; writing—original draft preparation, F.E.E.-R.; writing—review and editing, F.E.E.-R., A.G.-D., E.R., F.M.d.T. and G.M.-O.; visualization, F.E.E.-R., A.G.-D., E.R., J.C., F.M.d.T., F.C. and G.M.-O.; project administration, G.M.-O.; funding acquisition, G.M.-O. and F.E.E.-R. All authors have read and agreed to the published version of the manuscript.

**Funding:** Project RTI2018-101085-R-C31 (MINORVIN) was funded by MCIN/AEI/10.13039/501100011033 and by ERDF 'A Way of Making Europe'. F.E.E-R. received a grant (PRE2019-089073) funded by MCIN/AEI/10.13039/501100011033 and ESF 'Investing in Your Future'.

**Data Availability Statement:** The original contributions presented in the study are included in the article, further inquiries can be directed to the corresponding author.

**Conflicts of Interest:** The authors declare no conflicts of interest. The funders had no role in the design of the study; in the collection, analyses, or interpretation of data; in the writing of the manuscript; or in the decision to publish the results.

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
