# Peer review of "Phenological Evaluation of Minority Grape Varieties in the Wine Region of Madrid as a Strategy for Adaptation to Climate Change"

_horticulturae, doi:10.3390/horticulturae10040353_

Round 1

Reviewer 1 Report

Comments and Suggestions for Authors

In this manuscript authors investigate the changes in the meteorological conditions in the previous for years (2020-2023) in comparison to a long dataset (1957-2023) for the same location (Alcala de Henares, Spain), and the phenological characteristics and basic grape juice composition of 34 Spanish grapevine varieties during the same four years period (2020-2023) in the context of problems related to climate change and the possible introduction in a wider production of some minor Spanish grapevine varieties.

Although the thematic of the manuscript regarding the climate change as a current major problem in viticulture worldwide, and it surely has an importance for the region where the investigation was carried out, most of the text of this manuscript should be thoroughly rewritten and the data should be elaborated differently, if aimed to be published in Horticulturae journal.

In its current version it is written at a low standard for scientific publication, the text is not fluent, some wrong interpretations are given, the data should be differently elaborated and presented (and discussed), and most of the text should be carefully rewritten, taking care of every detail. Moreover, a similar paper by the same authors was published previously (https://www.mdpi.com/2311-7524/8/11/984 ) and the illogical presentation of results (details are given below) may raise a suspicion that it was aimed to make this manuscript different from the previous publication, although very similar data were used.

Specific comments:

Line 2: In the title instead of Madrid use the name of the wine region.

Lines 18-22: This text should be thoroughly rewritten.

Lines 23-25: These data are presented only for veraison.

Lines 25-27: These data are not presented in the manuscript (but only mentioned briefly in the text).

Line 26: Most probably you referred here to titratable acidity, expressed as g/L of tartaric acid.  

Line 26: Please check the value of pH, 5.5 is too high.

Lines 46-47: The reason of earlier bud break are higher temperature during winter, not the change in late frosts.

Lines 55-56: Illogical sentence.

Line 113: Change the word ‘campaigns’ to ‘seasons’ (here and in the rest of the manuscript).

Lines 145-146: First question: What do you mean by ‘the duration of phenological stages’? If it is the period from the onset of a given stage to the onset of the following stage, then this should be stated appropriately.

Lines 145-146: Second question: Why you presented the data of the duration of these specific stages in the manuscript? It does not make a lot of sense of it, because these data are mostly a response of the meteorological condition during a specific phenological stage, and for the season. In other words, for a given season, it does not make a big impact if flowering lasted seven or ten days. On the other hand, it is extremely important when the onset of some particular phenological stage occurred (day of year), and how long was the period between two stages, but these data are not presented in this manuscript.

Line 165: For the historical series the data from 1957 to 2019 should be used (and not until 2023).

Line 203: Even or seven?

Line 205: What do you mean by a term ‘les intense winter’?

Figure 1. a): Why do you not present the historical data for minimum and maximum temperature?

Line 218: What do you mean by a term ‘milder’?

Line 221: From June to August the data are above average.

Lines 225-226: Make it clearer.

Line 247: Do not use the term ‘between’ if the comparison was not made.

Line 248: The difference is length is among seasons?

Line 251: Change ‘later’ to ‘longer’.

Lines 252-255: These data are not presented in tables or figures, so they should be added or the text should be deleted.

Lines 256-258: These data are not presented in tables or figures.

Table 2: Write ‘Days’ instead of ‘Ds’.

Table 2 (and the corresponding text): There is no sense to asses the GDD during the duration of these specific phenological stages, as it is logical and highly expected that the longer duration will result with higher GDD.

Lines 268-272: Please use the English abbreviations, not Spanish.

Lines 267-276: These data are not presented in tables or figures.

Line 280: The accumulation of the sugars indicates the beginning of the ripening process, whereas the color change follow later.

Lines 280-285: This is most probably not the cause of this.

Line 287: It is not clear what does it mean ‘according to veraison’? According to the onset of veraison, or?

Figure 2: The grouping of the varieties seems quite arbitrary. Which method you use to make the groups?

Figure 2: Delete year 2019.

Figure 2: Please state what is a Day 0?

Line 299: As stated above, there is no sense to make the correlation of GDDs and Ds of these specific phenological stages, as it is logical and highly expected that the longer duration will result with higher GDD.

Line 315: Change ‘second’ to ‘last’.

Line 321: Change ‘Quality’ to ‘Composition’.

Line 323: Where it is presented this classification of the varieties? Which variety belongs to which cluster?

Table 4: It does not make sense to present the data in this way, as it is not clear which variety belongs to which cluster, and also it is not clear the days (period) of maturation of these clusters.

Lines 349-351: This is a contrasting expression.

Lines 353-355: These data are not presented in tables or figures.

Discussion and Conclusions: The text should be thoroughly rewritten based on the changes made to the presentation of the data, as indicted above.

Final comment: Beside the part of the text which was commented above, in the manuscript it should be carefully rewritten most of the text.

Comments on the Quality of English Language

The text of this manuscript should be thoroughly rewritten by paying attention to mistakes in writing and the way of expression in English language.

Reviewer 2 Report

Comments and Suggestions for Authors

Review of the mauscript entitled „Phenological evaluation of minority grape varieties in Madrid as a strategy for adaptation to climate change”

The peer-reviewed manuscript addresses the response in the phenology and grape quality of thirty-four Iberian native grape varieties to climate change in the Madrid area.

Climate change is currently one of the most frequently discussed topics in viticulture research. The article takes into account the needs of the wine sector related to the changing climate and weather anomalies, which significantly affect the development of vines and the quality of wine raw materials. The authors present the response of Iberian native wine varieties to weather conditions in four consecutive growing seasons. Based on the results obtained, they indicate varieties whose cultivation minimizes the negative impact of weather and at the same time allows maintaining the identity of the region and preserving the style of the wines originating from it. The research methodology, the method of processing meteorological data, parameters regarding the phenological phases of vines and grape quality, and the statistical analysis of the results are correct and generally accepted in research related to the present topic. The article brings new cognitive and practical issue. It is an important contribution to research on viticulture and counteracting the negative consequences of climate change. After minor corrections related to style, it may be accepted for further stages of the editorial process.

 Deatailed comments

Minor stylistic corrections are indicated in the text, e.g.

l. 200-201. „Monthly absolute minimum and maximum temperatures became more extreme and tended to be warmer”.

 l. 203-204. „Average temperatures between 3-5 °C warmer...”

The temperature cannot be defined as warmer. It may be higher or larger. For example, the air  is warmer.

Date of manuscript received: 13 March 2024

Date of this review: 15 March 2024

Reviewer 3 Report

Comments and Suggestions for Authors

The contribution entitled Phenological evaluation of minority grape varieties in Madrid (Alcalá de Henares) as a strategy for adaptation to climate change is focused on current topics in the field of viticulture. The contribution includes a comprehensive evaluation of a total of 34 Spanish minority varieties grown in the region of Spain. Individual varieties were evaluated in relation to the course of phenological stages (bud break, bloom, veraison, and maturity) and complete cycle.

The introduction of the contribution is prepared at a good level, the team of authors focuses mainly on the evaluation of temperature conditions, which can influence the phenophase of the vine to the greatest extent. In this context, significant temperature fluctuations that occur as a result of climate change can cause a number of complications for growers in the cultivation and production of wine. It is essential to know how temperature influences both the reproductive cycle and vegetative development and to observe differences between evaluated varieties. The goal of the work is appropriately and clearly formulated.

The methodological part is processed in a logical sequence, including the statistical evaluation methods used.

The results part is processed to a sufficient extent and is appropriately divided into individual sub-chapters. I propose to supplement the axis labels in Figure 2 (x axis). This graph describes very well the distribution of evaluated varieties according to earliness. I have a comment about chapter 4. Discussion, where the obtained results are commented on, but they are compared to the works of other authors to a relatively small extent. At the end of the contribution, I would recommend adding a proposal for the implementation of experiments in the next period. After incorporating these comments, the text can be accepted for publication.

Comments on the Quality of English Language

 Minor editing of English language required

Round 2

Reviewer 1 Report

Comments and Suggestions for Authors

The authors have revised the manuscript according to the given comments and I suggest to accept it for a publication.